# Multi-Robot Formation Control Based on CVT Algorithm and Health Optimization Management

Kai Cao [1,2,3], Yangquan Chen [1,2,3], Song Gao [1,2,*], Hang Zhang [2] and Haixin Dang [2]

1  School of Mechatronic Engineering, Xi'an Technological University, Xi'an 710021, China; caokai@xatu.edu.cn (K.C.); ychen53@ucmerced.edu (Y.C.)
2  School of Electronic Information Engineering, Xi'an Technological University, Xi'an 710021, China; ckstory2020@gmail.com (H.Z.); dx002978@gmail.com (H.D.)
3  School of Engineering (MESA-Lab), University of California, Merced, CA 95343, USA
*  Correspondence: gaos@xatu.edu.cn; Tel.: +86-029-86173350

**Abstract:** In view of the low formation redundancy in the traditional rigid formation algorithm and its difficulty in dynamically adapting to the external environment, this study considers the use of the CVT (centroidal Voronoi tessellation) algorithm to control multiple robots to form the desired formation. This method significantly increases the complexity of the multi-robot system, its structural redundancy, and its internal carrying capacity. First, we used the CVT algorithm to complete the Voronoi division of the global map, and then changed the centroid position of the Voronoi cell by adjusting the density function. When the algorithm converged, it could ensure that the position of the generated point was the centroid of each Voronoi cell and control the robot to track the position of the generated point to form the desired formation. The use of traditional formations requires less consideration of the impact of the actual environment on the health of robots, the overall mission performance of the formation, and the future reliability. We propose a health optimization management algorithm based on minor changes to the original framework to minimize the health loss of robots and reduce the impact of environmental restrictions on formation sites, thereby improving the robustness of the formation system. Simulation and robot formation experiments proved that the CVT algorithm could control the robots to quickly generate formations, easily switch formations dynamically, and solve the formation maintenance problem in obstacle scenarios. Furthermore, the health optimization management algorithm could maximize the life of unhealthy robots, making the formation more robust when performing tasks in different scenarios.

**Keywords:** CVT (centroidal Voronoi tessellation); multi-robot; formation control; obstacle avoidance; health optimization management





## 1. Introduction

With the continuous development of wireless communication technology, multi-robot systems have attracted widespread attention in the field of intelligent control and have been used in scenarios such as resource exploration, forest fire monitoring, map construction, and disaster rescue. Multi-robot formation control [1–3], as an important branch of multi-intelligence systems, requires multiple robots to maintain a certain formation and avoid collisions while performing tasks, while minimizing the impact of environmental constraints on the multi-robots and influencing them to complete a specific task. Over the years, a large number of research results have been produced for the multi-robot formation control problem, and a variety of formation control methods represented by the follower method, the behavior-based control method, the virtual structure method, and the artificial potential field method have been proposed. These formation methods have their advantages and disadvantages. For different application scenarios, the above methods are often used in combination to achieve the desired formation effect.

The following leader method is one of the earliest solutions for multi-robot formation control. In [4,5], leader selection is described as an optimization problem; it adjusts the position of the follower based on the position of the leader, forming the desired formation. To maintain the required formation shape of spacecraft, Chen Qifeng set up virtual spring dampers between the spacecraft and realized a formation control method based on the artificial potential field [6]. In [7,8], based on the principle of the virtual structure method, the formation control problem is described as a synchronous control problem, and a decentralized trajectory tracking controller was designed with the position error as feedback information to complete the functions of describing behavior and assigning tasks. In the above-mentioned formation control method, task-oriented, efficient cooperation between robots is used to improve the efficiency in completing tasks. In order to be more in line with the actual environment and enhance the robustness and anti-interference ability of the system, JF Flores Resendiz specified the communication diagram between robots to make the input speed constrain multiple mobile robots to converge to the desired formation, which improved the overall obstacle avoidance performance of the system. The amount of feedback was used as the control, based on the differences in neighbor behavior, thereby reducing communication loss and computing cost [9]. Zhou used a distributed control protocol to solve the problem of multi-mobile-robot formation considering sampling data and time delay [10]. Yao solved the formation control problem of a heterogeneous multi-robot system under the premise of considering the robot kinematics uncertainty and communication delay, and used the Lyapunov function to prove the stability of the adaptive sliding mode controller [11]. Taking into account the complexity of the robot's task execution scenarios, Qiu Huaxin designed a distributed formation optimization control framework for UAVs based on the hierarchical learning behavior of a pigeon flock, which transformed the multi-objective optimization problem into a single UAV, solving the multi-objective optimization problem to achieve stable formation flying in a complex obstacle environment [12].

In summary, controlling multiple robots to maintain a fixed formation can result in completing the specified tasks more efficiently, and the formation control should be extended to a more efficient dynamic mode, that is, provided the robots maintain the corresponding formation as much as possible, to reduce the rigidity of the formation. Therefore, the question of how to achieve the formation with less energy consumption in a shorter time has also received considerable attention. Chen Yangquan used the distribution and neutralization of multi-robot clusters in the diffusion process of toxic substances as the background, and combined the CVT algorithm with multi-robot control for the first time [13]. Cortes proposed a discrete-time-based CVT algorithm that can use Euclidean metrics to achieve coverage of non-convex environments [14]. The algorithm was proved by solving the coverage of obstacles and exploration problems in non-convex indoor environments. In [15], a coverage algorithm is defined that is subject to differential constraints in a non-convex environment. The algorithm introduces a limited range of sub-divisions to limit the amount of communication between robots, and the convergence of the algorithm is verified in simulation. In [16], an adaptive control algorithm is designed to drive the robot to the weighted centroid of the corresponding Voronoi cell, and finally the Lyapunov function is used to prove that the robot network can converge to the optimal configuration.

In the process of using the formation to complete the task, the completion of the overall task and the health of the robot are equally important. In past formation use, the user seldom considered the impact of various harsh environments on the health of the robot under real conditions. The impact of such environments may degrade the health of the robot or even cause it to fail to work, which will have a significant impact on the completion of the task and the performance of the robot [17]. Therefore, in an intelligent and efficient formation control system, these abnormal possibilities must be considered and the individual situation must be assessed, in order to maximize the formation performance and mission completion, and to achieve a higher level of overall mission performance than non-health-perception systems.

In terms of the impact of individual health-degrading tasks, [18] proposed a task allocation system based on health management which can predict the impact of various abnormalities on the task status through dynamic programming and select measures to mitigate these effects. The authors in [19–21] investigated the impact of different robot driving capabilities in the team on the coverage task, using online and distributed methods to learn the changes in the relative driving performances of the robots, which can be a trade-off between time and the energy consumption of the robot, to ensure that the robots converged to the best configuration.

In view of the above problems, an efficient and dynamic multi-robot formation algorithm should be developed that can form an efficient dynamic formation with the least cost and make flexible changes according to the obstacle environment. Most importantly, the impact of the environment on the health of robots must be taken into account, in order to be able to make real-time dynamic adjustments to the robots and to improve the performance and robustness of the robot formation.

On this basis, this study combines the CVT (centroidal Voronoi tessellation) algorithm with multi-robot formation control. First, we used the CVT algorithm to complete the Voronoi division of the task map and designed the controller to control the robot to track the specified Voronoi cell centroid. When the CVT algorithm converged, it was guaranteed that the robots formed the specified desired formation in the specified area according to the optimal distribution. Secondly, considering that there are obstacles in the real environments where robots perform tasks, this study also verified the formation maintenance experiment in an obstacle environment. Under the premise of ensuring the uniform distribution of the global density function, the density value corresponding to the obstacle position was reduced to ensure that the centroid was not divided into the obstacle area. That is, the characteristics of the CVT algorithm were used to realize the repulsive force field that drives the robot to avoid obstacles. It was necessary to set up additional potential field functions to complete the collision avoidance and obstacle avoidance control of formation robots, which significantly improved the flexibility and scalability of the algorithm. Finally, the health optimization management algorithm was used to maximize the endurance of unhealthy robots under the premise of minor changes to the original framework, so that the system could deal better with unforeseen damage such as mechanical failures and emergencies, making the formation more robust when performing tasks in different scenarios.

The rest of this article is organized as follows. Section 2 provides the CVT-based multi-robot formation control algorithm and cost function optimization process, and gives a proof of the stability of the algorithm. Section 3 introduces the health optimization management algorithm used in this study and the insertion construction method used when the robot's health is exhausted. Section 4 presents the numerical multi-robot formation simulation and health optimization management optimization under the control of the algorithm presented in this paper, to verify the effectiveness of the algorithm. In Section 5, the robot platform is verified in a laboratory scenario to illustrate the practicability of the algorithm presented in this paper. Section 6 presents the conclusions and discusses future work.

## 2. Multi-Robot Formation Control

### 2.1. Problem Description

The aim of the Voronoi tessellation algorithm is to divide the area containing $n$ generation points into $n$ cells, where each cell contains a generation point and other points in the cell are closest to the generation point of the cell, to ensure that the cost function of the system reaches the minimum value. Centroid Voronoi tessellation is a special case of Voronoi tessellation. It sets the density function of the target area, and its generation point is the centroid of each Voronoi cell. Firstly, we assume that n virtual wireless sensors with limited energy are set in the target area. To maximize the service life of the sensor, we consider a solution where the sensor only interacts with the nearest robot, which can effectively reduce the power consumption for information transmission and obtain the surrounding environment information faster.

It is assumed that a single robot in a multi-robot system has a position $p_i \epsilon Q, i \epsilon \{1, 2, \ldots, n\}$, $n$ robots move in a bounded interval, and $Q \in S^2$, $S^2$ is a two-dimensional grid map of the target area. The position of the wireless sensor is expressed as $q_i \epsilon S^2, i \epsilon \{1, 2, \ldots, n\}$. Taking the robot position as the generation point of the Voronoi division, the Voronoi cell $V = \{V_1, V_2, \ldots, V_n\}$ in region $Q$ is expressed as in Equation (1):

$$V_i = \{q \in Q | \|q - p_i\| \le \|q - p_j\|, \forall j \ne i\} \tag{1}$$

If the generation point in the Voronoi cell is equal to the centroid of the Voronoi area, it is called the centroid Voronoi partition (CVT). Figure 1a,b shows the Voronoi division and centroids, respectively, where the crosses in Figure 1a represent the Voronoi cell centroids and the points and irregularities are shown in Figure 1a,b. The polygons represent the spawning point and the corresponding Voronoi cell, respectively.

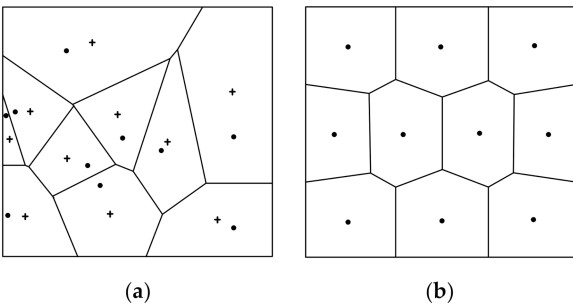

(**a**)  (**b**)

**Figure 1.** Schematic diagram of Voronoi cell: (**a**) VT; (**b**) CVT.

Let $\rho(q)$ be the density function defined in the target region $Q$. A larger $\rho(q)$ value means a higher density of certain attributes or a more special location, which is usually measured by the sensor network in order to represent the attributes of some areas. The cost function in the bounded region $Q$ is defined as in Equation (2):

$$E(p) = \sum_{i=1}^{n} E^i = \sum_{i=1}^{n} \int_{V_i} \|q - p_i\|^2 \rho(q) dq \tag{2}$$

The number of wireless sensors is usually limited and discontinuous. Assuming that the sensor network is evenly distributed in the target area $S^2$, the cost function in the bounded area $Q$ can be expressed as in Equation (3):

$$E(p) = \sum_{i=1}^{n} E^i = \sum_{i=1}^{n} \sum_{j \in V_i} \|q - p_i\|^2 \rho(q) \tag{3}$$

In order to ensure that multiple robots maintain a reasonable formation during the task, this study emphasizes that the formation can reduce the overall energy loss. Therefore, we transform the formation problem into the tracking problem for the Voronoi interval centroid, and make the robots maintain Voronoi cells with similar quality through the adjustment of the density function. The mass of the cells can be expressed in a continuous version and a discrete version, as shown in Equations (4) and (5), respectively.

$$M_{V_i} = \int_{V_i} \rho(q) dq \tag{4}$$

$$M_{V_i} = \sum_{j \in V_i} \rho(q_j) \tag{5}$$

Assume that the average mass of all cells in area $Q$ is $\overline{M}$. The difference between the Voronoi cell mass and the average mass is expressed as the difference. When the difference

reaches the minimum, the mass of each cell is approximately equal by default, that is, the workloads of the robots are approximately equal. The formula for the Voronoi mass difference is shown in Equation (6):

$$E_2(V) = \sum_{i=1}^{n} E_2^i = \sum_{i=1}^{n} \left\| M_{V_i} - \overline{M} \right\|^2 \tag{6}$$

In summary, in the process of performing multi-robot system tasks, keeping $E(p)$ and $E_2(V)$ to a minimum can minimize the energy loss of information interactions and maintain the target formation, that is, reasonable formation control can be transformed into a multi-objective optimization problem.

### 2.2. Cost Function Optimization

In most application scenarios, the CVT algorithm is used to minimize the cost function of the system. The mathematical formula of the cost function is shown in Equations (7) and (8):

$$E_V(p) = \int_{V_i} |q - p_i|^2 \rho(q) dq, i = 1, 2, \ldots, n \tag{7}$$

$$E_V(p) = \sum_{V_i} \sum_{j=1}^{N} |q_j - p_i|^2 \rho(q_j) \tag{8}$$

The so-called cost function can also be expressed as the variance or energy function. The cost function estimates the error between the generated point and the centroid of the Voronoi cell through the density function of the region. The proof of minimizing the cost function using the CVT algorithm is as follows:

1.  The position change of the generated point $p_j$ is shown in Equation (9):

$$E_{V_j}(p_j + \varepsilon) - E_{V_i}(p_j) = \int_{V_j} \rho(q)\varepsilon \{\varepsilon + 2(p_j - q)\} dq \tag{9}$$

2.  Simplifying Equation (9) to Equation (10):

$$\frac{E_{V_j}(p_j + \varepsilon) - E_{V_i}(p_j)}{\varepsilon} = \int_{V_j} \rho(q)\varepsilon \{\varepsilon + 2(p_j - q)\} dq \tag{10}$$

3.  Assuming that $\varepsilon$ in Equation (10) tends to 0, we obtain:

$$\dot{E}_{V_j}(p_j) = 2\int_{V_j} \rho(q)(p_j - q) dq \tag{11}$$

4.  Deforming Equation (11) into Equation (12):

$$\dot{E}_{V_j}(p_j) = 2p_j \int_{V_j} \rho(q) dq - 2\int_{V_j} q\rho(q) dq \tag{12}$$

5.  If the derivative part is 0, as shown in Equation (13), the solution at this time is the minimum solution.

$$p_j{}^* = \frac{\int_{V_j} q\rho(q) dq}{\int_{V_j} \rho(q) dq} \tag{13}$$

At this time, the position of the generated point is the centroid position of the Voronoi cell, and hence the central Voronoi tessellation can minimize the cost function of the system. Applying the CVT algorithm to the multi-robot system can minimize the cost function. Taking the formation task as an example, multi-robot systems are required to form a specific formation and maintain a fixed distance between robots. Therefore, the cost function is composed of the error between the actual distance and the expected distance, as shown in Equation (14):

$$E(p) = \sum_{i=1}^{n} \sum_{j=N_i} \frac{1}{2} \left( \|p_i - p_j\| - d_{ij} \right)^2 = \sum_{i=1}^{n} E_i \left( \|p_i - p_j\| \right) \tag{14}$$

where $d_{ij}$ is the set threshold distance between robots $i$ and $j$ and the immediately expected distance. The distance between the current robots and the expected formation is measured by the cost function $E$. When the contemporary price function is 0, the robot forms the expected formation, and the derivative of $E_i(x)$ is calculated as shown in Equation (15):

$$\frac{\partial E_i}{\partial p_i} = \sum_{j \in N_i} \frac{\|p_i - p_j\| - d_{ij}}{\|p_i - p_j\|} \left( p_i - p_j \right)^{\mathrm{T}} \tag{15}$$

Equation (15) shows that if the distance between two robots is less than the set threshold distance, the weight value is negative and the distance between the two robots is controlled to increase. When the distance between robots is greater than the set threshold distance, the weight $\omega_{ij}$ will control the distance between two machines to shorten. When the set threshold distance $d_{ij} = 0$, the task will be transformed into a group consistency problem. The task performed by the multi-robot system is transformed into the optimization problem of the cost function. The constraint-based formulation derivation can enhance the adaptability of the system, and the distribution of the controller can be guaranteed by selecting different weights or defining some assumptions. The robots only need to execute their inputs to ensure the convergence of the system within a limited time.

### 2.3. Formation Control Algorithm

The CVT algorithm is usually used to solve the optimal distribution of static resources. If the iterative speed of the CVT evolution process is slower than that of the motion planning algorithm, the algorithm can be used for motion control of multiple robots. Through the static or dynamic communication topology between robots, the multi-robot formation uses a consistency algorithm to drive robots to achieve a consistent state. The desired formation of the robots is composed of the centroids of the Voronoi region divided by the CVT algorithm. The purpose of the consistency algorithm is to drive all robots to their corresponding centroids.

Assuming that $p_i = (x_i, y_i)$ is the actual position of the robot and $p_i^* = (x_i^*, y_i^*)$ is the desired position of the robot, the robot is driven to the desired position via Equation (16):

$$p_i^* = \frac{\sum\limits_{j=1}^{n} g_{ij} q_j}{\sum\limits_{j=1}^{n} g_{ij}} \tag{16}$$

where $n$ is the number of robots and $g_{ij}$ is the parameter in the robot communication topology matrix. The Algorithm 1 flow chart is shown in Algorithm 1.

### 2.4. System Stability Analysis

Taking the differential car as an example, this study assumes that the wheeled robot is a rigid body that can rotate freely. Its simplified motion model is shown in Figure 2.

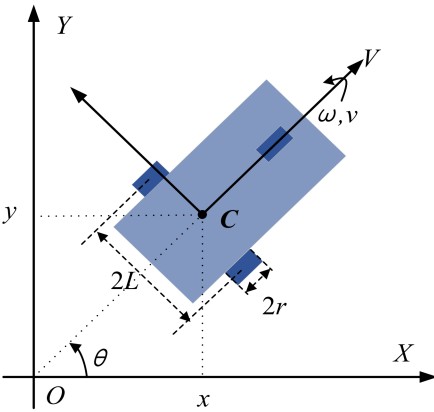

**Figure 2.** Robot kinematics model.

---

**Algorithm 1:** The multi-robot formation control Algorithm 1

---

**Require:**
The robot performs SLAM mapping and positioning;
Set the initial area density function $\rho(q)$.
**Procedure:**
  1:   **While** $((p_i - p_i^*) \geq \varepsilon)\&\&\partial(E(p)) \neq 0$
  2:   Construct a Delaunay triangulation that contains the initial positions of all robots;
        Traverse the triangle list;
  3:   **if** Triangles $b,c,d$ have common sides with the current triangle $a$ **then**
  4:   Connect the outer centers of triangles $b,c,d$ to the outer centers of triangles $a$;
  5:   **else**
  6:   Calculate the outermost midline of the triangle;
  7:   **end if**
  8:   Store in Voronoi side list;
  9:   Construct Voronoi graph grid of region $Q$;
 10:   Drive the robot to reach the corresponding centroid;
 11:   **End While**

---

Set $C(x, y)$ as the coordinates of the wheeled robot, and $\theta$ as the forward direction angle. The state information of the robot is represented by $P = (x, y, \theta)^{\mathrm{T}}$ and $v$ and $w$ are the linear velocity and angular velocity of the robot, while $2L$ is the length between two wheels and $2r$ is the diameter of the wheel. The kinematic state-space equation can be obtained as shown in Equation (17):

$$\begin{cases} \dot{x} = v\cos(\theta) \\ \dot{y} = v\sin(\theta) \\ \dot{\theta} = \frac{v\tan(\varphi)}{2L} \end{cases} \tag{17}$$

According to the robot kinematics model and the control law, it is necessary to maintain zero dynamic control input, $\dot{p}_i = 0$, that is, $u_i = 0$. The ideal output result is that the robot reaches the centroid position of CVT convergence. Considering that the problem is a tracking problem from the robot position to the CVT centroid position, the Lyapunov function was used to prove the convergence of the system.

In the proof process, the integral part of the control law is separated to obtain Equation (18):

$$\dot{p}_i = u_i = k_p(p_i - p_i^*) - k_d \dot{p}_i \tag{18}$$

The Lyapunov function is defined as in Equation (19):

$$V(p_i) = \frac{1}{2}k_p\|p_i - p_i^*\| + \frac{1}{2}p_i^2 \tag{19}$$

The Lyapunov function is derived to obtain Equation (20):

$$\dot{V}(p_i) = k_p(p_i - p_i^*)\dot{p}_i + \dot{p}_i\ddot{p}_i \tag{20}$$

Replacing the parameter $\ddot{p}_i = 0$ in Equation (20) with $\dot{p}_i$ to simplify, gives Equation (21):

$$\dot{V}(p_i) = -k_d \dot{p}_i^2 \leq 0 \tag{21}$$

Therefore, it can be concluded that the Lyapunov function is a semi-negative definite function. As shown in Equation (21), when the time tends to infinity the system is asymptotically stable over a large range, if and only if the input is 0, i.e., $\dot{V}(p_i) = 0$. According to the LaSalle principle, if the surface fine diversity is limited the robot will gradually converge to the centroid of the specified Voronoi cell.

$$u_i = 0 = k_p(p_i - p_i^*) \tag{22}$$

## 3. Health Optimization Management

*Problem Description*

To improve the health of each robot and enhance the robustness and task completion of the formation, health degradation should be considered. Ensuring the survivability of a single robot will often have a significant impact on the task performance and future reliability of the overall formation, as factors such as low battery, high power consumption rate or unforeseen damage to physical structures can cause the robot to slow down in speed or task processing. Although these health problems caused by degraded operating conditions or malfunctions do not need to be dealt with immediately, continued operation will have a significant impact on the mission performance and future reliability of the robot itself and even on the formation.

In response to the above problems, the formation health optimization management algorithm was introduced to restrict the speed of the unhealthy robot's Voronoi cell, in order to make it move more conservatively, maximizing the endurance of the unhealthy robot. This makes the formation system better able to deal with unforeseen damage such as mechanical failures and emergencies, making it more robust when performing tasks in different scenarios.

In this algorithm, $u_i$ is the control input of robot $i$. To better describe the health status of the robot, in response to the above problems, $h_i(t)$ is specially introduced, where $h_i(t)$ at time $t$ can be expressed as:

$$\begin{cases} h_i(t) = h_i(0) - d_i(t) \\ d_i(t) = \int_0^t k_{d,i} u_i^2 dt + f_{d,i}(t) \\ \dot{h}_i(t) \leq 0, \dot{f}_{d,i}(t) \geq 0, h_i(t), f_{d,i}(t) \in [0,1] \end{cases} \tag{23}$$

where $h_i(0)$ is the initial health of the robot, $d_i(t)$ is the health loss of the robot at each moment, $k_{d,i}$ is the usage gain coefficient, and $f_{d,i}(t)$ is the negative impact of the external environment on the robot, such as in damage caused by external collisions. Assuming that the health of each robot decreases with the progress of its motion control and possible external damage, Equation (23) is a function of its control accumulation and any external damage that may be encountered. Controlling the accumulation means that fast-moving and turning robots lose their health faster than slow-moving or stationary robots, which can be simplified to a combination of remaining power and structural health in a physical sense.

The current health status of a multi-robot formation can be calculated using the average health level of all robots:

$$\begin{cases} h^*(t) = \frac{1}{n} \sum_{i=1}^{n} h_i(t) \\ \dot{h}^*(t) < 0, h^*(t) \in [0,1] \end{cases} \tag{24}$$

Assume that each robot can obtain the overall average health level $h^*(t)$ of all robots in the entire task area by communicating with neighboring robots within a limited range. The movement of each robot is controlled by a proportional controller, which is controlled by the proportional derivative controller of the formation health optimization management algorithm:

$$\begin{cases} u_i(t) = k_p \cdot [1 + k_{h,i}(t)] \cdot (p_i - p_i{}^*) \\ e_i(t) = h_i(t) - h^*(t) \\ k_{h,i}(t) = k_{h_p} e_i(t) + k_{h_d} \dot{e}_i(t) \end{cases} \tag{25}$$

where $k_{h,i}(t)$ represents the gain obtained by the robot changing the mobile formation health optimization management system, $e_i(t)$ represents the difference between a robot and the overall average health level $k_{h_p}$, and $k_{h_d}$ represents the gain coefficient of the formation health optimization management algorithm.

Finally, combining the optimization of robot health management and the flexibility of the CVT algorithm, the remaining robots re-divide the original area after the robot whose health value is reduced to zero stops working (the robot cannot continue to work normally due to energy exhaustion, failure, or the deterioration of the external environment), to minimize the impact of this type of robot on the formation task.

Based on the traditional CVT construction algorithm, the insertion construction method was introduced, so that in the original CVT configuration, when the robot number is reduced, all the triangles whose circumscribed circle contains new points are recorded, the common edges that affect the construction of the new triangle are deleted, and then new edges are re-planned, until all points are inserted. The specific process is shown in Figure 3a–d.

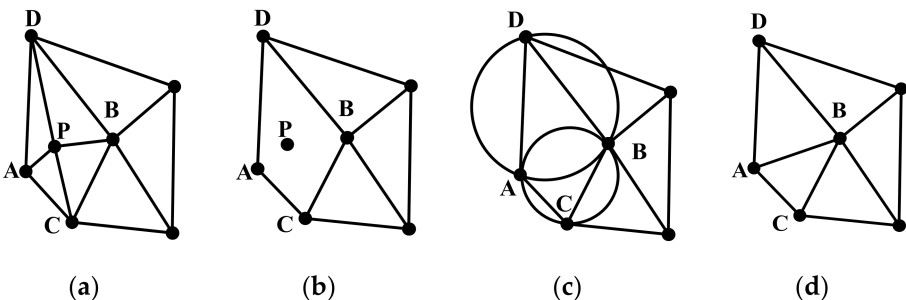

|           |           |           |           |
|-----------|-----------|-----------|-----------|
| (**a**)   | (**b**)   | (**c**)   | (**d**)   |

**Figure 3.** Insert construction method: (**a**) robot P has a health value of zero; (**b**) delete the public edge related to P; (**c**) re-plan neighboring robots; (**d**) form a new distribution.

The specific control Algorithm 2 steps in this study were as Algorithm 2.

---

**Algorithm 2:** CVT-based multi-robot formation control and health optimization management Algorithm 2

---

**Require:**
  The initial position of a group of robots $P : \{p_i\}_{i=1}^n, h_i(t), h_i(0)$;
  Boundary information $\partial Q$ of task area $Q$, area density function $\rho(q)$;
**Procedure:**
  1:   Calculate the centroid of the Voronoi cell $\{p_i^*\}_{i=1}^n$ and $\{V_i\}_{i=1}^n$ of the robot position $P$;
  2:   **While** $((p_i - p_i{}^*) \geq \varepsilon)$&&$\partial(E_V(p)) \neq 0$
  3:     Calculate $d_i(t)$ and $h_i(t)$ for each robot
  4:     **if** $h_i(t) \leq 0$ **then**
  5:      Use the insert construction method to re-form and divide the area
  6:     **end if**
  7:     Calculate $h^*(t)$ and $u_i(t)$ for the entire formation
  8:     Drive the robot to the center-of-mass position $p_i = p_i{}^*, (i = 1, \ldots, n)$;
  9:     Generate a new set of robot positions $P : \{p_i = p_i^*\}_{i=1}^n$;
  10:    Calculate the new centroid $\{p_i^*\}_{i=1}^n$ and $\{V_i\}_{i=1}^n$ of the robot position set $P$;
  11:  **End While**

---

When the formation is moving, the entire formation is recalculated to obtain the distribution in the environment. While controlling the robot to have the smallest possible impact on other robots, making the robots in the formation respond better to changes in the environment and improving the robustness and task execution ability of the formation in various environments.

## 4. Experiment and Simulation

### 4.1. Multi-Robot Formation Control Simulation Experiment

The simulation experiment in this article was performed using the MATLAB simulation platform under the Windows operating system. In MATLAB, assuming that the robot was a first-order dynamic model, the size of the designated area was $10 \times 10 \text{ m}^2$, and it was planned to control four robots in this area to complete square, diamond and linear formations. A total of $1000 \times 1000$ virtual sensors were evenly placed in the area, and the values of the sensors were fitted to the density function of the CVT algorithm. By modifying the density function, the robots were controlled to form the desired formation. It can be seen from Figure 4 that the initialization positions of the four robots were all near the origin.

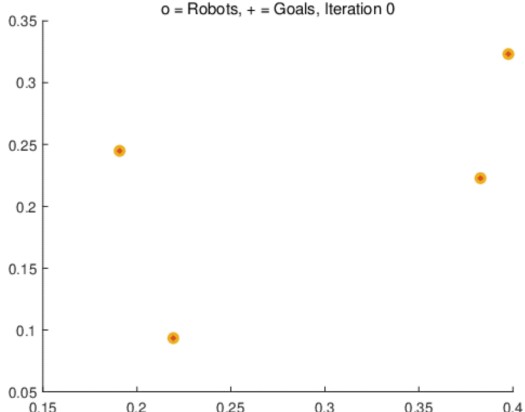

**Figure 4.** Initial positions of robots.

4.1.1. Square Formation Simulation and Results Analysis

First, we initialized the density function to a constant value of 1. As shown in Figure 5a, when the initial density function in the area $Q$ is constant, the Voronoi algorithm is used to divide the area, and the robot tracks the position of the generated point to complete the square formation. In Figure 5b, the solid lines in different colors represent the tracking motion trajectories of the robots.

In the initial stage of the experiment, the positions of the robots were randomly distributed near the origin, and the cost function value was $6 \times 10^3$ at this time. With the iteration of the CVT algorithm, the generation point tends to the centroid of each Voronoi cell, and the cost function value gradually decreases. When the number of iterations $k = 35$, the controller is affected by the integral saturation and overshoots, and the cost function increases to $2.8 \times 10^3$. When $k = 55$ or so, by reducing the integral control effect of the controller or separating the integral control process, the cost function value is gradually reduced. When $k = 90$, the generation points of the Voronoi cells reach their respective centroid positions, and the CVT algorithm converges, as shown in Figure 5d. Figure 5c shows the position error between the robot and its corresponding centroid. In the initial stage of control, the error between the robot and the generating point is the largest. When the CVT algorithm converges, the position error between the robot and the generating point eventually tends to zero.

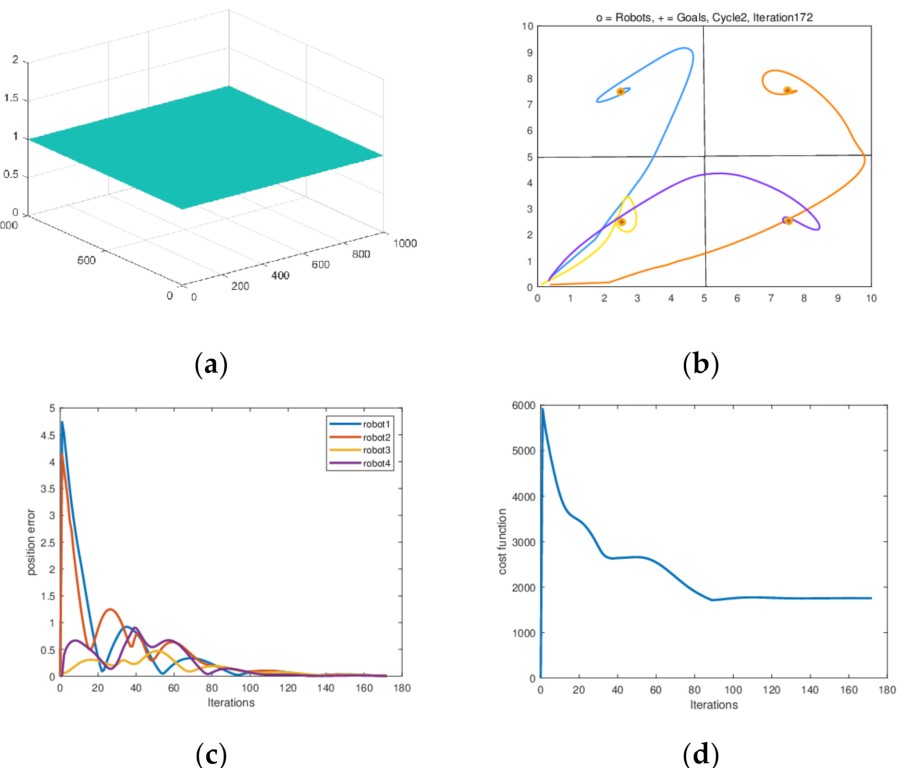

**Figure 5.** Multi-robot square formation simulation: (**a**) density function distribution; (**b**) square formation; (**c**) position error; (**d**) cost function.

#### 4.1.2. Simulation and Results Analysis for the Diamond Formation

Based on the density function being a constant, by adjusting the density function to a Gaussian function, the diamond formation of the robot can be completed. The Gaussian function equation is shown in Equation (26). The density function distribution is shown in Figure 6a, and the formed diamond formation is shown in Figure 6b.

Since the robot first needs to complete the Voronoi division under the constant density function distribution, the starting positions of the robots were the same as for the square formation. The cost function value at the initial moment was $6.2 \times 10^3$. When the number of iterations of the CVT algorithm $k = 30$, the controller is affected by the integral saturation to produce overshoot, and the cost function increases to $3.3 \times 10^3$. When the number of iterations is about 90, the generation points of the Voronoi cells reach their respective centroid positions, and the CVT algorithm converges under the constant density function distribution. After adjusting the density function to a Gaussian distribution, the CVT algorithm finally converged again when the number of iterations $k = 200$. The cost function is shown in Figure 6d. Figure 6c shows the position error between the robot and its corresponding centroid. When the number of iterations $k = 140$, the position of the center of mass changes due to the switching of the density function. It can be seen that the position error between the robot and the corresponding generating point has increased significantly. When the CVT algorithm reconverges, the position error tends to zero again.

$$\sigma(x, y) = \exp^{-\delta(a(x-x_c)^2 + b(y-y_c)^2)} \tag{26}$$

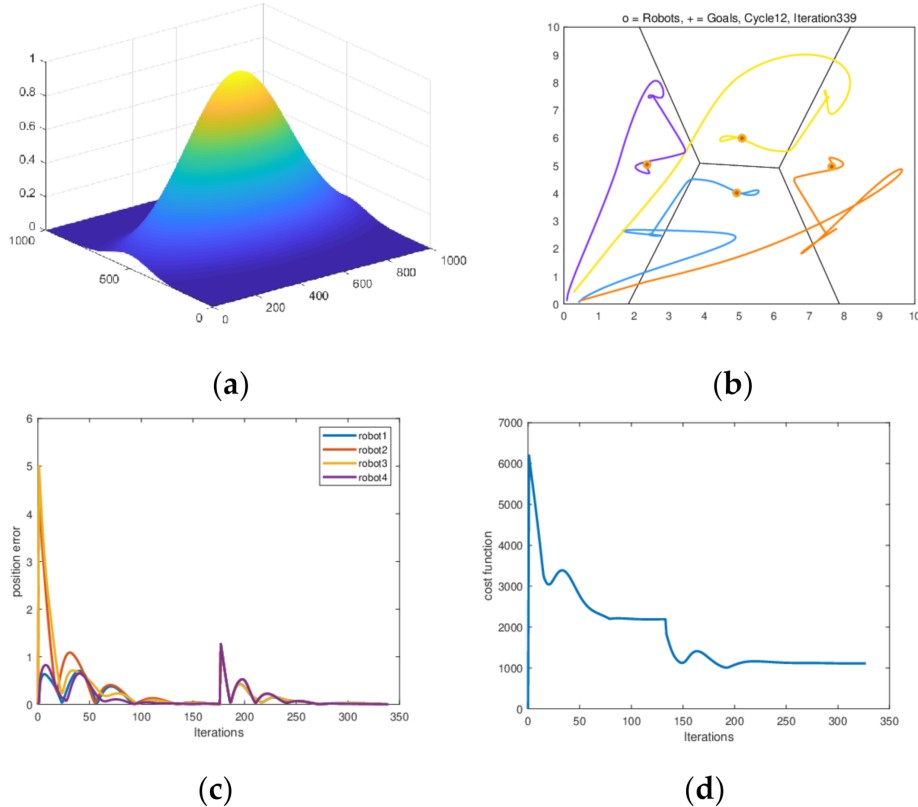

**Figure 6.** Multi-robot diamond formation simulation: (**a**) density function distribution; (**b**) diamond formation; (**c**) position error; (**d**) cost function.

### 4.1.3. Straight Line Formation Simulation and Results Analysis

On the basis that the density function is a constant, by adjusting the density function to a V-shaped function, the linear formation of the robot can be completed. The V-shaped function equation is given in Equation (27). The density function distribution is shown in Figure 7a, and the resulting linear formation is shown in Figure 7b.

In the linear formation process, the robot still needs to complete the Voronoi division under the constant density function distribution first. In the iterative process of the CVT algorithm, the controller is affected by the integral saturation to produce an overshoot. When the number of iterations $k$ = 30 to 40, the cost function increases from $3.45 \times 10^3$ to $3.65 \times 10^3$. After separating the integral control process, the controller tends to a stable state again, and the cost function value decreases again. When $k$ = 140, the cost function is $3.5 \times 10^3$, and the CVT algorithm converges under the constant density function distribution. At this time, the density function was adjusted to a V-shaped function, and the optimal solution was reached again when the number of iterations $k$ = 180, as shown in Figure 7d. In Figure 7c, when the number of iterations $k$ = 140, the position of the center of mass changes due to the switching of the density function. It can be seen that the position error between the robot and the corresponding generating point has increased significantly. When the CVT algorithm reconverges, the position error tends to zero.

$$\sigma(x, y) = \exp^{-\delta(ax+by+c)^2} \tag{27}$$

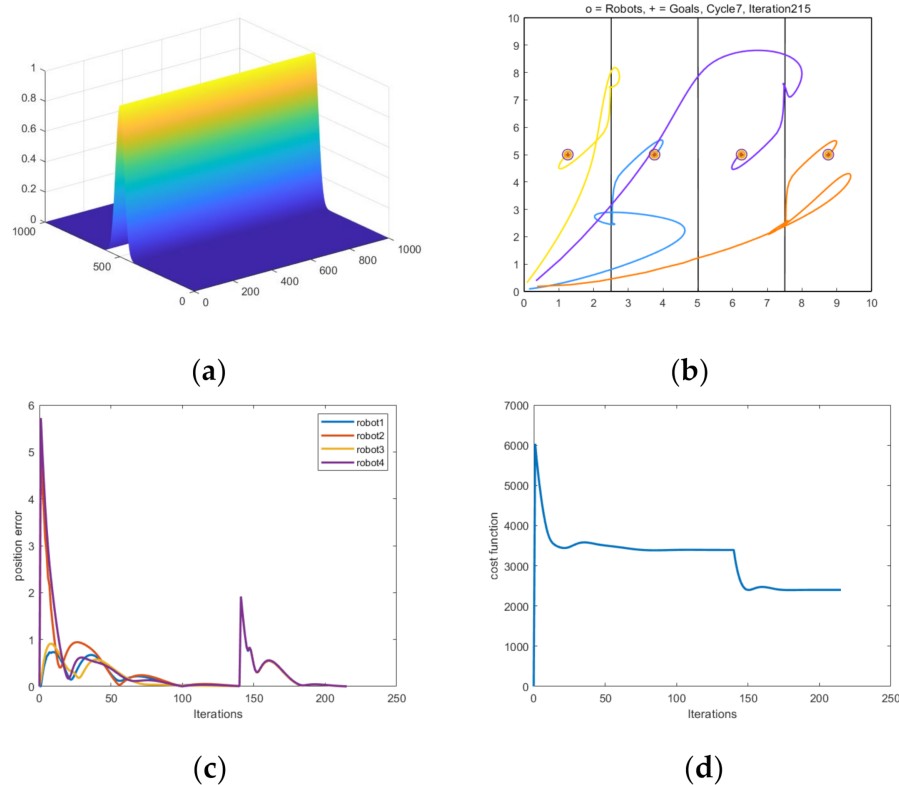

**Figure 7.** Multi-robot linear formation simulation: (**a**) density function distribution; (**b**) linear formation; (**c**) position error; (**d**) cost function.

### 4.1.4. Simulation and Results Analysis of Dynamic Formation in Barrier-Free Scenes

In a real scene, there are obstacles in the environment where the robots perform tasks, so in this study we also verified the formation-maintaining algorithm for multiple robots in an obstacle scenario. Under the premise of ensuring the uniform distribution of the global density function, we first used a specific constraint shape to specify the desired formation and then reduced the density value corresponding to the obstacle's position to ensure that the center of mass was not divided in the obstacle area. That is, the characteristics of the CVT algorithm were used to realize the repulsive force field that drives the robot to avoid obstacles, thereby realizing a multi-robot formation in an obstacle scene.

In MATLAB, it was also assumed that the robot was a first-order dynamic model, the formation scenario map was $30 \times 30$ m$^2$, $3000 \times 3000$ virtual sensors were evenly placed on the map, and the values of the sensors were fitted to the density function of the CVT algorithm. In the process of dynamic formation, we consider constructing virtual constraint boundaries. Since the density function is always uniformly distributed, the robots are distributed in a square formation in the area $Q$, surrounded by the dynamic virtual constraint boundary.

The density in the scene map was set to be uniformly distributed, and the area $Q$, surrounded by the virtual constraint boundary, was evenly divided into four Voronoi cells using the Voronoi algorithm. Assuming that the movement path of area $Q$ is known, the control robot completes the tracking of its corresponding center of mass and maintains a square formation. The formation under the obstacle-free scenario is shown in Figure 8a,b; when the number of iterations $k = 75$, the position error eventually tends to zero. To measure the effect of formation maintenance, we assume that the geometric centroid of the Voronoi algorithm is the center of the formation. The sum of the distances between the robots and the formation center is shown in Figure 8d, and when the distance tends to $\sqrt{2} * l$, the formation reaches a stable state, where $l$ is the side length of area $Q$.

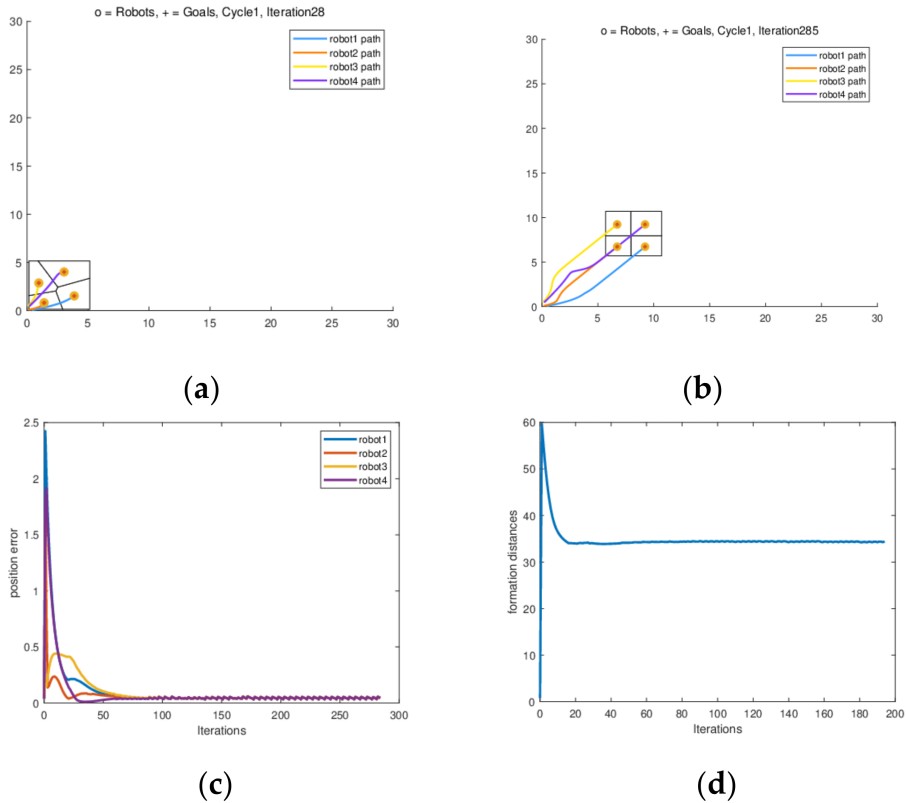

**Figure 8.** Dynamic formation simulation under the obstacle-free scenario: (**a**) $k = 28$; (**b**) $k = 285$; (**c**) position error; (**d**) formation preservation.

4.1.5. Dynamic Formation Simulation and Results Analysis under Random Obstacle Scenarios

When there are obstacles in the scene, by reducing the density of the obstacle area the Voronoi centroid can be divided into obstacle-free areas, so that the robot can avoid obstacles in the process of tracking the centroid. The density distribution is shown in Figure 9a. Figure 9b–e shows the multi-robot formation control under the obstacle map at different moments; when the number of iterations $k$ is between 10 and 125, the position of the generating point must be constantly changed to avoid obstacles in the path. The error also changes accordingly. When the multi-robot system passes through the obstacle area, that is, when $k = 130$, the robots quickly restore the initial formation. Under the action of the PID controller, the error fluctuation gradually tends to zero. The basis for judging whether the formation is stable is whether the sum of the distances between the robots and the formation center tends to $\sqrt{2} * l$, as shown in Figure 9g. When the number of iterations $k = 130$, the formation-maintaining function value remains at 8.4, that is, the formation returns to a stable state.

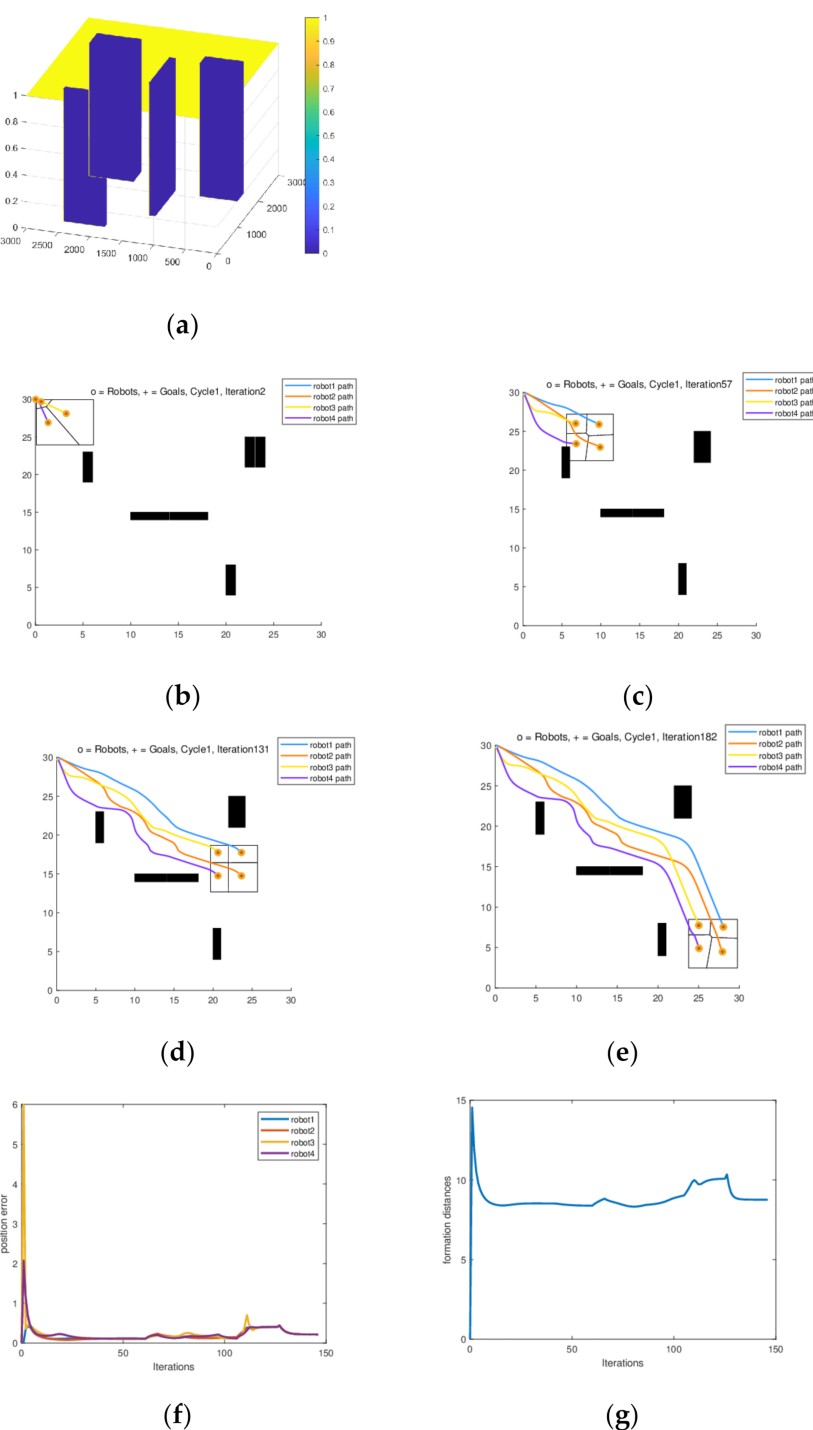

**Figure 9.** Dynamic formation simulation under random obstacle scenario: (**a**) distribution of density function under random obstacle scenario; (**b**) $k = 2$; (**c**) $k = 57$; (**d**) $k = 31$; (**e**) $k = 182$; (**f**) position error; (**g**) formation preservation function.

### 4.1.6. Simulation and Results Analysis of Dynamic Formation in Narrow Passage

Assuming that the obstacles in the scene are large and sparsely distributed, as shown in Figure 10a, we realize obstacle avoidance by setting the density of the obstacle area to 0 and then dynamically zooming the boundaries of the Voronoi division area to drive the overall formation to shrink to pass through the narrow-passage terrain. Figure 10f,g shows the position error and formation maintenance of the system, respectively. When the sum of the distances between the robots and the center of mass is equal to $\sqrt{a^2 + b^2}$, where a and b are the length and width of the region $Q$, respectively, the formation remains stable. When

the number of iterations reaches *k* = about 60, the side length of the area *Q* is contracted, and the robot can track the center of mass more easily. When the four robots maintain a linear formation up to *k* = 350, all the robots pass through the narrow passage. Due to the change of the density function of the position of the robot at this time, to form the optimal configuration, the position error of the robot fluctuates slightly. At *k* = 350~400, to form a square formation from a linear formation, there is an adjustment process before the regional boundary is adjusted to a square, as shown in Figure 10g. When the number of iterations *k* = 580, the robots formed the expected square formation, and the position error of the robots also tended to zero.

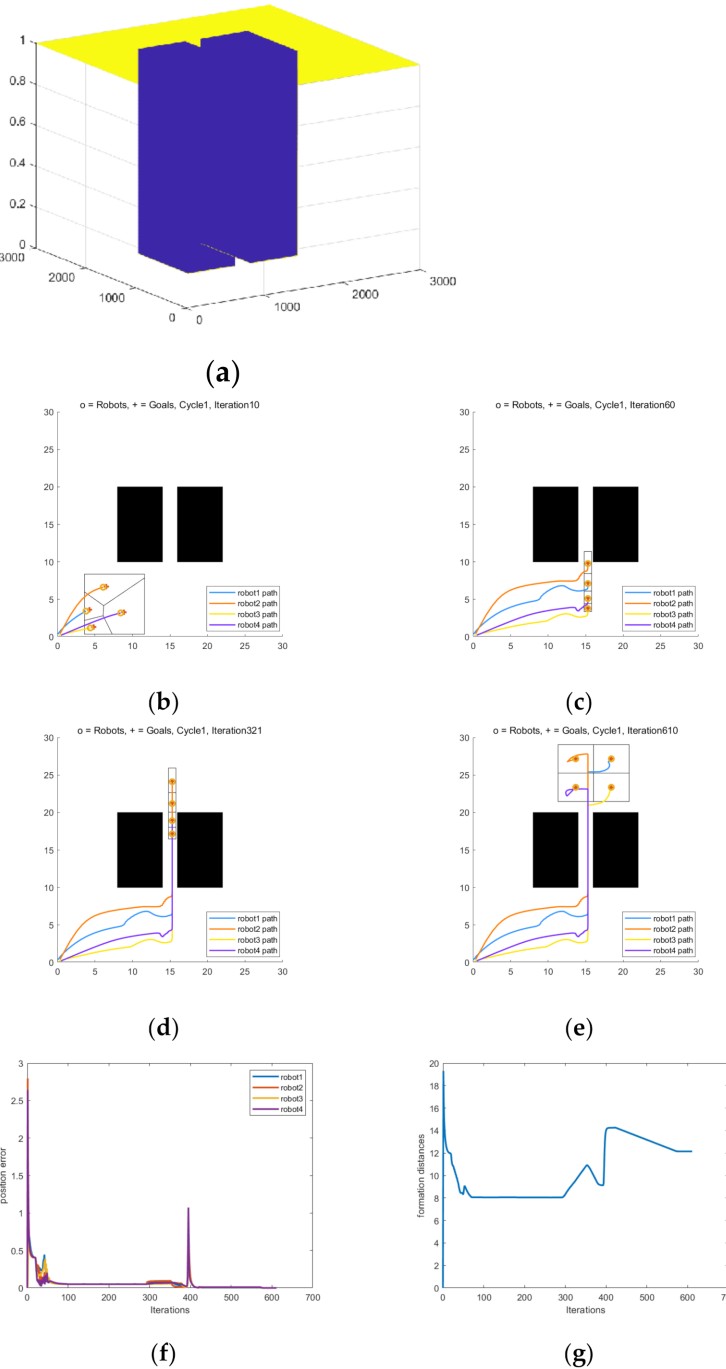

**Figure 10.** Dynamic formation simulation in a narrow channel: (**a**) density function distribution in a narrow channel; (**b**) *k* = 10; (**c**) *k* = 60; (**d**) *k* = 321; (**e**) *k* = 610; (**f**) position error; (**g**) formation preservation function.

### 4.2. Simulation of Formation Health Optimization Management

The performance of the health optimization management algorithm can be assessed via the health loss value when the robot group reaches the optimal configuration. In the implementation of health optimization management algorithms, the goal is to maintain the health of all robots as much as possible and reduce disproportionate health degradation. This article proposes and studies four scenarios:

① None of the robots in the group have any abnormal health behaviors;
② The health degradation rate of a single robot increases;
③ The health value of a single robot at the beginning of the task is lower than other members of the group;
④ The health of a single robot decreases suddenly.

These scenarios represent some common expected health problems. Scenario ① represents the best-case scenario, in which no robot suffers from abnormal health behaviors, so the health optimization management algorithm is used to optimize the performance of each robot. Scenario ② may occur during a period of engine failure, reducing power efficiency. Scenario ③ may indicate that the battery is not fully charged at the beginning of the task. Scenario ④ may be a small component failure, such as a structural fracture. Under these circumstances, the deterioration of the operational status or failure does not require immediate recovery, but continued operation may affect the overall future operational status and performance of the formation.

In these four scenarios, the four robots form a square formation as an example. During the formation, the health degradations described in the above four scenarios may be encountered. For each scenario, two simulations were performed: one for normal operation and the other for health optimization management.

#### 4.2.1. All Robots Are Healthy

Figure 11a,b describes the results of the first case, where each robot is initialized with the same health level, and the health of each robot decreases normally with its movement. Figure 11a depicts the health of each robot during formation control without health optimization management. When all the robots reach the final configuration, the health of robot 1 drops more than the rest, reaching the final configuration when $h_i(t) = 0.26$. In Figure 11b, with optimized health management, the same robot extends its lifespan and does not reach the final configuration until $h_i(t) = 0.48$, increasing the lifespan by 22%.

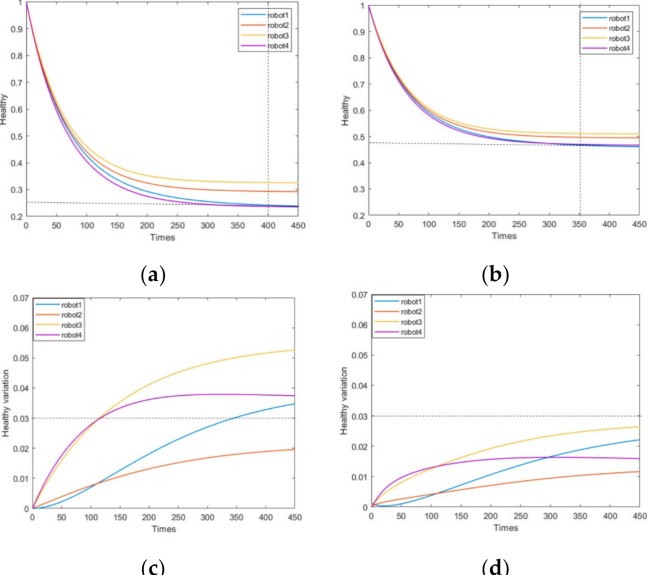

**Figure 11.** Comparison before and after adding HOM in the first scenario: (**a**) $h_i(t)$ of the robot under normal conditions; (**b**) $h_i(t)$ of the robot under HOM; (**c**) $e_i(t)$ under normal conditions; (**d**) $e_i(t)$ under HOM.

It can be seen from Figure 11c that if there is no health optimization management, the differences in robot health will diverge over time, while in Figure 11d, the differences are bounded.

4.2.2. The Initial Power of a Single Robot Is Lower than That of Other Robots, Resulting in a Decrease in Battery Life

Figure 12a,b describes the results in the second case, where each robot is initialized with the same health level. Figure 12a depicts the health of each robot during formation control without health optimization management. When all the robots reach the final configuration, the health of robot 4 drops more than the rest, reaching the final configuration when $h_i(t) = 0.04$. In Figure 12b, with optimized health management, the same robot extends its lifespan and does not reach the final configuration until $h_i(t) = 0.38$, increasing the lifespan by 34%.

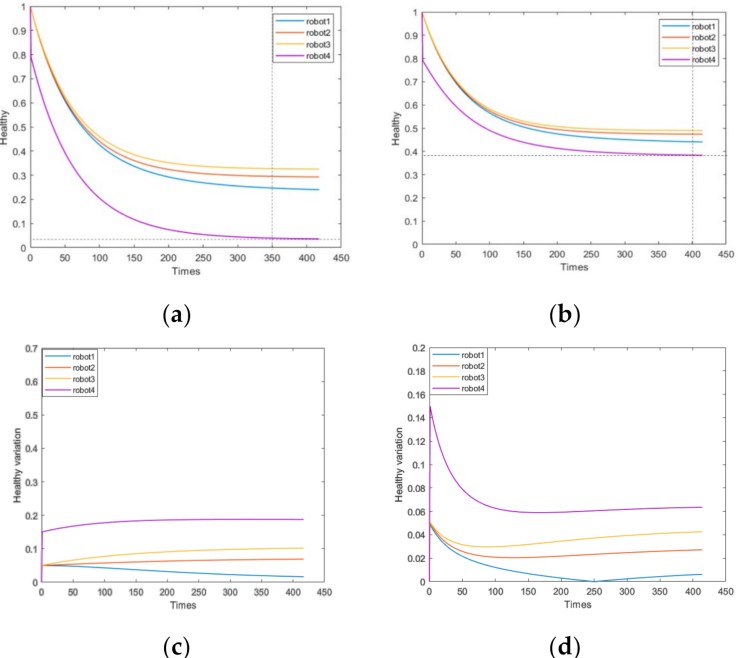

**Figure 12.** Comparison before and after adding HOM in the second scenario: (**a**) $h_i(t)$ of the robot under normal conditions; (**b**) $h_i(t)$ of the robot under HOM; (**c**) $e_i(t)$ under normal conditions; (**d**) $e_i(t)$ under HOM.

As can be seen from Figure 12c, if there is no health optimization management, the differences in robot health diverge greatly with the passage of time, while in Figure 12d, the difference is small.

4.2.3. The Failure of a Single Robot Component Caused by the External Environment, Such as a Damaged Wheel That Causes a Sharp Deterioration in the Health of the Robot

Figure 13a,b describes the results of the third case, where each robot is initialized with the same health level. Figure 13a depicts the health of each robot during formation control without health optimization management. When all the robots reach the final configuration, the health of robot 4 drops more than the rest, reaching the final configuration when $h_i(t) = 0.15$. In Figure 13b, with optimized health management, the same robot extends its lifespan and does not reach the final configuration until $h_i(t) = 0.4$, increasing the lifespan by 25%.

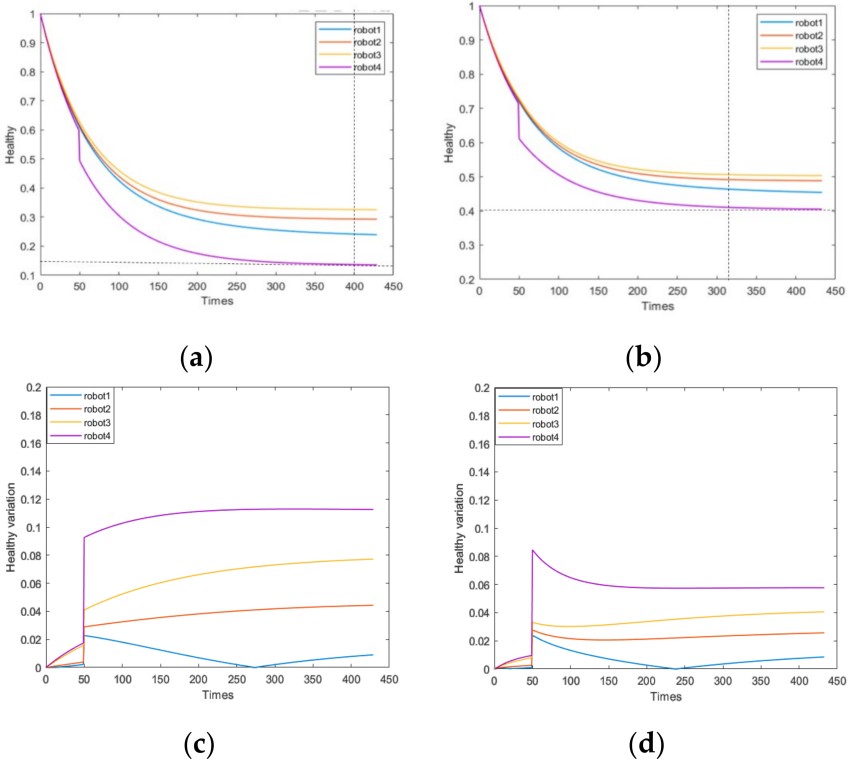

**Figure 13.** Comparison before and after adding HOM in the third scenario: (**a**) $h_i(t)$ of robots under normal conditions; (**b**) $h_i(t)$ of robots under HOM; (**c**) $e_i(t)$ under normal situations; (**d**) $e_i(t)$ under HOM.

As can be seen from Figure 13c, if there is no health optimization management, the differences in robot health diverge with time and are large compared with the average value, while in Figure 13d, the differences are small.

#### 4.2.4. Due to the Abnormal Fault of the Motor of a Single Robot, the Power Consumption Rate of the Robot Increases

Figure 14a,b describes the results of the fourth case, in which each robot is initialized with the same health level. Figure 14a depicts the health of each robot during formation control without health optimization management. When all robots reach the final configuration, the health of robot 4 decreases more than the rest, reaching the final configuration at $h_i(t) = 0.02$. In Figure 14b, with health optimization management, the same robot extends its life and does not reach the final configuration until $h_i(t) = 0.3$, increasing its life by 28%.

As can be seen from Figure 13c, if there is no health optimization management the differences in robot health will diverge greatly with the passage of time, while in Figure 13d, the differences are small.

In the above four cases, the health optimization management greatly increases the lifetime of the weakest robot, and the adjustment of the existing framework is very small. Health optimization management is very important for multi-robot formations to ensure that the formation can operate for a long period, to reduce health loss, and to ensure the reliability of robots in the task.

It can be seen from Table 1 that by applying intelligent health balancing to each robot, the overall health value can be improved by about 20~30%, and the whole robot formation is more robust to health changes caused by many problems that may reasonably be expected. The system can deal better with accidental damage caused by collision, mechanical failure, or insufficient maintenance.

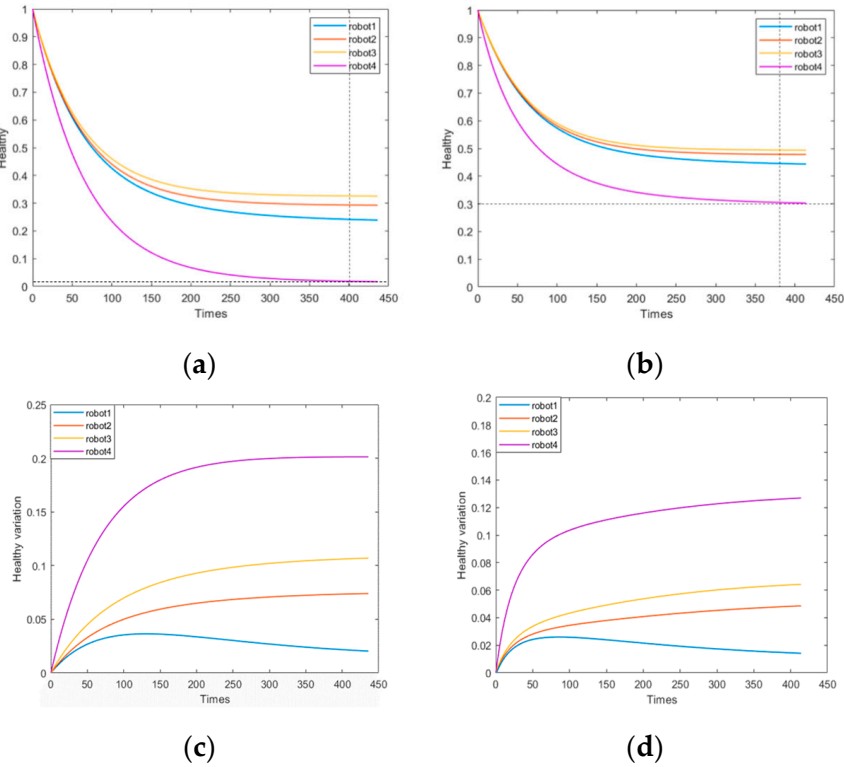

**Figure 14.** Comparison before and after adding HOM in the fourth scenario: (**a**) $h_i(t)$ of robots under normal conditions; (**b**) $h_i(t)$ of robots under HOM; (**c**) $e_i(t)$ under normal conditions; (**d**) $e_i(t)$ under HOM.

**Table 1.** Comparison of simulation results in four scenarios before and after adding the HOM algorithm.

|  | $h_i(t)$ When the System Is Stable (Normal Situation) | $h_i(t)$ When the System Is Stable (HOM) | Percentage Increase in System Health Value |
|---|---|---|---|
| **Scenario 1** | 0.26 | 0.48 | 22% |
| **Scenario 2** | 0.04 | 0.38 | 34% |
| **Scenario 3** | 0.15 | 0.40 | 25% |
| **Scenario 4** | 0.02 | 0.30 | 28% |

4.2.5. Formation Simulation and Results Analysis after Robot Stops Working

When $h_i(t) \leq 0$ due to other reasons, the robot is considered to be unable to work normally. In order not to affect the overall task of the formation, the insertion construction method was used to regroup multiple robots and re-divide the task area. Taking the square formation as an example, as shown in Figure 15a,b, four robots form a square formation before the number of iterations $k = 120$. After that, when the robot in the lower right corner cannot work normally due to a fault, and $h_i(t) \leq 0$, as shown in Figure 15e, the insertion construction method is used to make the other healthy robots adopt a new formation and reconstruct a new division of the new service area.

As shown in Figure 15c, when the number of iterations reaches 280, the remaining three robots adopt a new formation in the area and realize a new division of the area. The red plus sign in the figure is the final position of the faulty robot. The position error and cost function diagrams for this process are shown in Figure 15d,f.

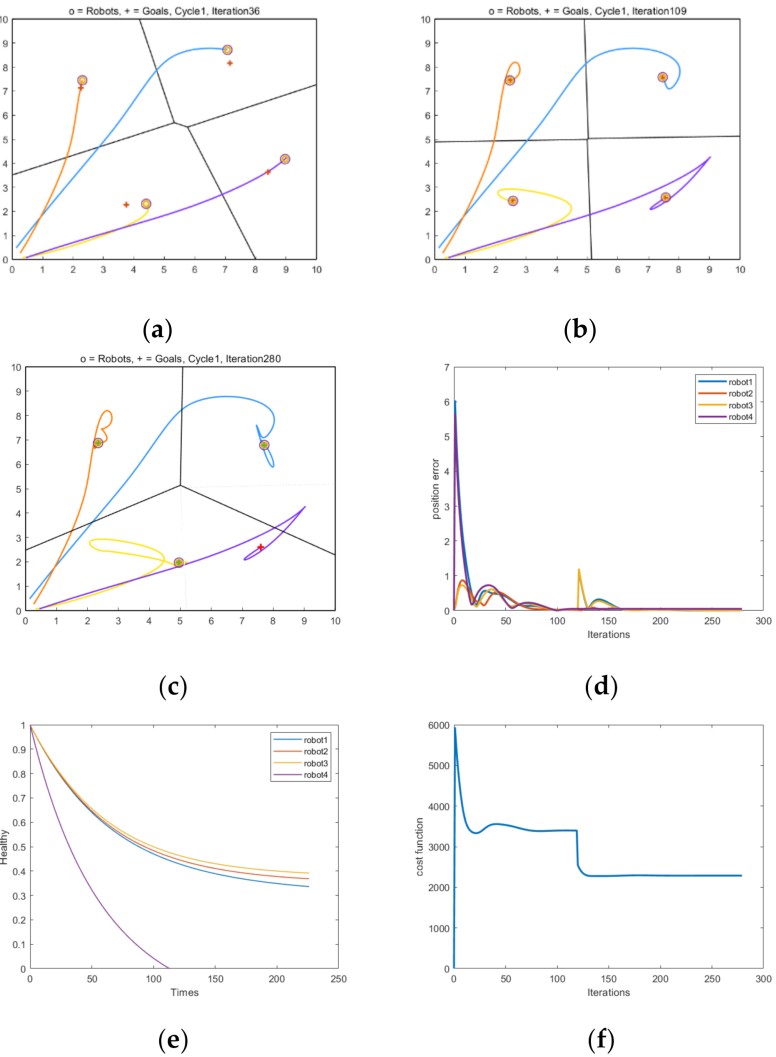

**Figure 15.** Robot formation after the introduction of the insert construction method: (**a**) $k = 36$; (**b**) $k = 109$; (**c**) $k = 280$; (**d**) position error; (**e**) $h_i(t)$ of each robot; (**f**) cost function.

As shown above, when the insertion construction method is introduced into the CVT it can not only reduce the impact on other robots when there is a faulty robot in the formation but can also compensate for the negative impact of the faulty robot as far as possible, so that the robots in the formation can better respond to the changes in the environment, and the formation control can better adapt to the actual situation, improving the execution and robustness of the robot formation.

## 5. ROS Robot Experiment

### 5.1. Multi-Robot Formation Experiment

The experiment in the real scene in this study was based on the TurtleBot3 robot, which is a mobile robot platform based on the ROS (robot operating system), with low cost and easy secondary development. Each TurtleBot3 robot can independently establish a local map of its surrounding environment through the gmapping function package provided by ROS, which provides SLAM for lidar, so that the robot can establish a grid-based 2D map according to the input and attitude lidar data. Taking the laboratory map as the designated area $Q$ of the robot formation, we set the condition that there were no obstacles in this area, as shown in Figure 16a. Four TurtleBot3 robots were randomly placed on the map. The robot completes its positioning using SLAM technology. The experimental results are shown in Figure 16b. The experiment was set so that when the Voronoi cost function reaches the optimal solution, the robot will reach the desired formation position under the

density function, that is, the target formation is formed. The difference from the simulation experiment is that the density function was not initialized to a constant in the initial stage of the experiment, and the target expected formation was determined first.

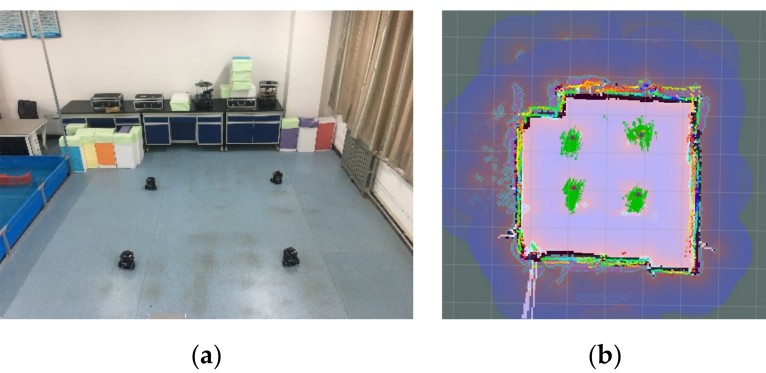

(**a**)　　　　　　　　　　　　(**b**)

**Figure 16.** Multi-robot formation experiment environment: (**a**) real scene; (**b**) RVIZ scene.

### 5.1.1. Square Formation Experiment

When the density function was set to a constant, the map was evenly divided into four generation points, as shown in Figure 17a. Figure 17c shows the initial positions of the robots. When the number of iterations $k = 200$, the CVT algorithm converges, as shown in Figure 17b. At this time, the cost function reaches the minimum value, and the robot completes the square formation on the map, as shown in Figure 17d.

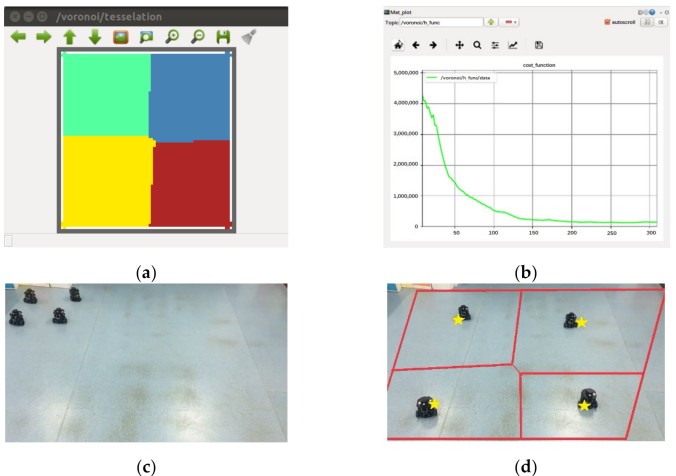

**Figure 17.** Square formation experiment: (**a**) density function distribution; (**b**) cost function; (**c**) initial position; (**d**) final position.

### 5.1.2. Diamond Formation Experiment

Figure 18a shows the Voronoi division under the Gaussian distribution density function, and Figure 18c shows the initial positions of the robots. Since the extreme point of the density function is in the central area of the map, the robots gradually approach the center of the map. Figure 18b shows the cost function when the density function is a Gaussian distribution. Compared with the uniform distribution, the initial value of the cost function is small, about $4.5 \times 10^2$, because the density value of most areas of the stage is approximately 0. When the CVT algorithm converges at iterations $k = 80$, the robot completes the diamond formation, as shown in Figure 18d.

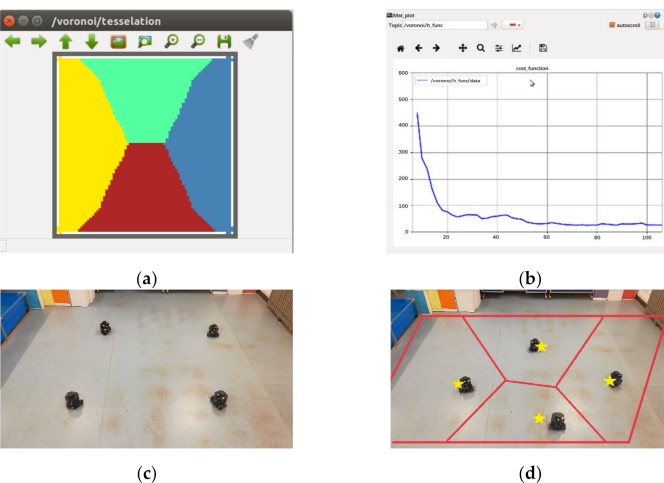

**Figure 18.** Diamond formation experiment: (**a**) density function distribution; (**b**) cost function; (**c**) initial position; (**d**) final position.

### 5.1.3. Linear Formation Experiment

Figure 19a shows the Voronoi division under the V-shaped density function, and Figure 19c shows the initial positions of the robots. Figure 19b shows the cost function when the density function is distributed in a V shape. The initial value of the cost function is about $6.5 \times 10^4$. When the CVT algorithm converges at iterations $k = 120$, the robots complete the linear formation, as shown in Figure 18d.

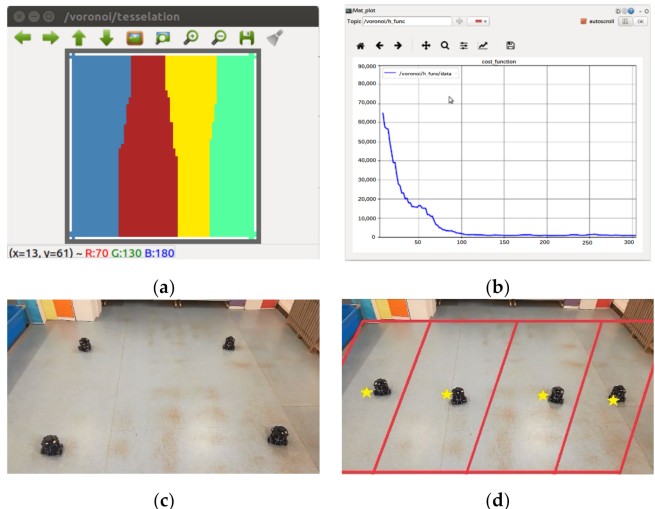

**Figure 19.** Linear formation experiment: (**a**) density function distribution; (**b**) cost function; (**c**) initial position; (**d**) final position.

### 5.2. Multi-Robot Dynamic Formation Experiment

For the dynamic formation scene, the algorithm was also verified using four TurtleBot3 robots. The experimental part mainly includes the dynamic formation in the obstacle-free scene, the dynamic formation in the random obstacle scene, and the formation transformation experiment. In the dynamic formation experiment in the obstacle-free scene and the random obstacle scene, the initial position of the robot is in the upper left corner of the map, as shown by the pentagram in Figure 20a. The target navigation point is located in the lower right corner of the map, at the red flag, and the black line is the trajectory of the multi-robot system. In the formation transformation experiment, the initial position of the robot is in the lower left corner of the map, shown as the pentagram position in Figure 20c. The target navigation point is located in the middle and upper part of the map, at the red flag position, and the black line in Figure 20c is the trajectory of the multi-robot system.

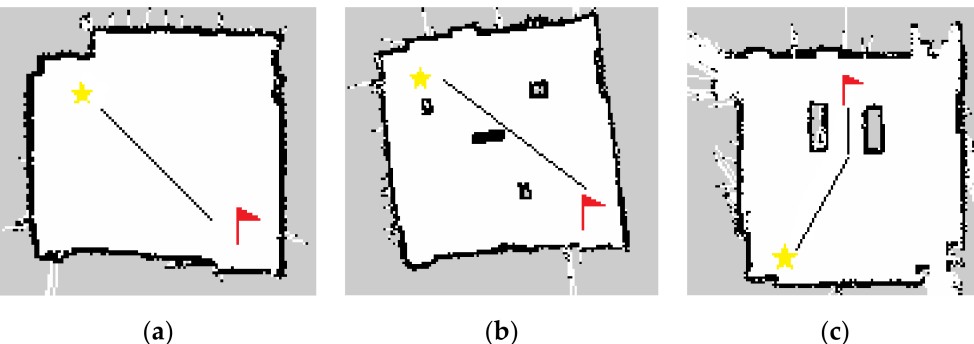

**Figure 20.** Experimental maps: (**a**) obstacle-free scene; (**b**) random obstacle scene; (**c**) narrow passage.

5.2.1. Dynamic Formation Experiment in Obstacle-Free Scene

In the obstacle-free scene, the robot is located in the upper left corner of the map at time $t_1$. When the Voronoi division algorithm converges, the robot reaches the centroid position of each Voronoi cell, as shown in Figure 21a. At this time, the constraint area $Q$ is driven to move in the specified direction and the robot is controlled to track the corresponding centroid position, as shown in Figure 21b,c. As shown in Figure 21d, the robot reaches the specified position at time $t_4$.

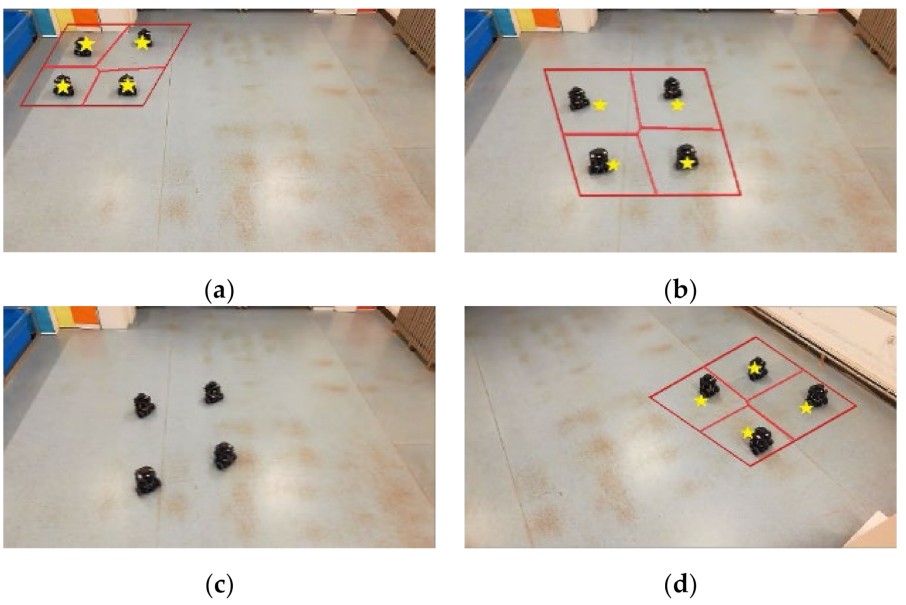

**Figure 21.** Multi-robot dynamic formation experiment in the obstacle-free scene: (**a**) $t_1 = 0$ s; (**b**) $t_2 = 64$ s; (**c**) $t_3 = 75$ s; (**d**) $t_4 = 120$ s.

5.2.2. Dynamic Formation Experiment in Random Obstacle Scene

In the scene with a random distribution of obstacles, similarly, the robot is located in the upper left corner of the map at time $t_1$. When the Voronoi division algorithm converges, the robot reaches the centroid position of each Voronoi cell, as shown in Figure 22a. At this time, the constraint area $Q$ is driven to move in the specified direction, taking the two-dimensional grid map information as the input and setting the area density value whose grid threshold is not 0 to 0, to ensure that the centroid will not be placed in the obstacle area, as shown in Figure 22b,c. As shown in Figure 22d, the robot reaches the specified position at time $t_4$.

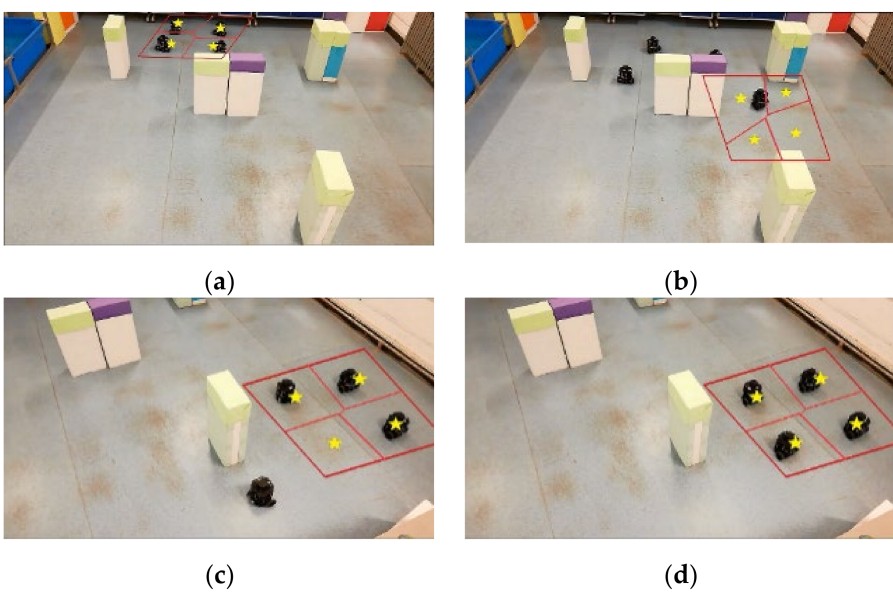

**Figure 22.** Multi-robot dynamic formation experiment with random obstacles: (**a**) $t_1 = 0$ s; (**b**) $t_2 = 32$ s; (**c**) $t_3 = 80$ s; (**d**) $t_4 = 102$ s.

### 5.2.3. Dynamic Formation Experiment in Narrow Channel Scene

In the narrow channel scenario, the formation needs to be changed to pass through the obstacles. The robot is located in the lower left corner of the map at time $t_1$. When the Voronoi division algorithm converges, the robot reaches the centroid position of each Voronoi cell, as shown in Figure 23a. At time $t_2$, we change the size of the constraint area $Q$ to transform the robot formation from a square to a straight line, as shown in Figure 23b,c. Finally, the robot formation reaches the designated position, as shown in Figure 23d.

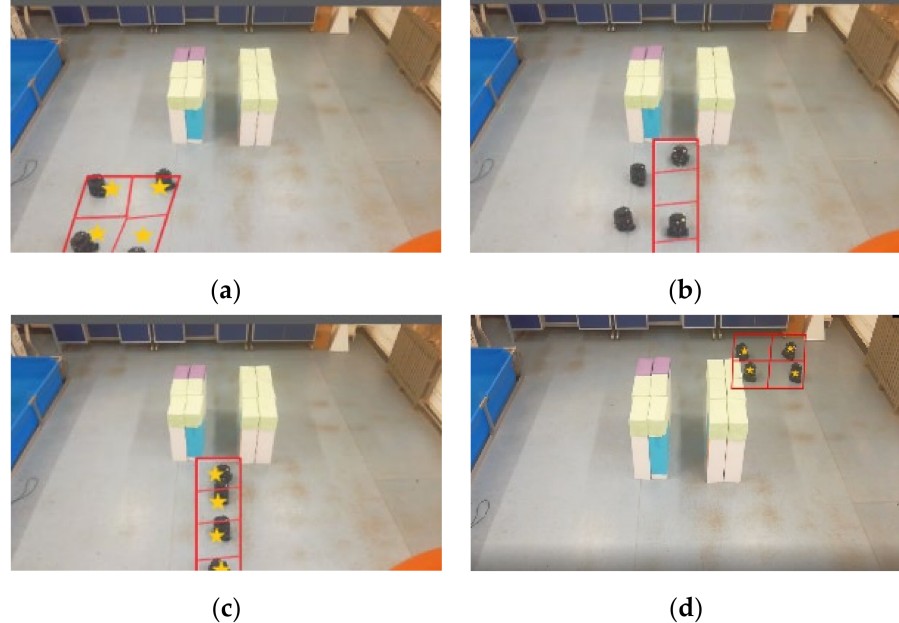

**Figure 23.** Multi-robot dynamic formation experiment in a narrow channel: (**a**) $t_1 = 0$ s; (**b**) $t_2 = 48$ s; (**c**) $t_3 = 61$ s; (**d**) $t_4 = 207$ s.

## 6. Summary and Outlook

This paper proposes a multi-robot formation control and health optimization management method based on the CVT algorithm, which has outstanding advantages compared with the traditional multi-robot formation algorithm.

(1) The algorithm converges one robot to the center of mass of each Voronoi unit, which avoids collisions between robots to the greatest extent, making the multi-robots form the desired formation at the specified position according to the optimal distribution and proving the related theories.

(2) Taking into account the existence of obstacles in the environment, by reducing the value of the density function of the obstacle position the robot can avoid the obstacle, improving the collision avoidance ability of the robot.

(3) The health optimization management algorithm was used to maximize the endurance of unhealthy robots and improve the stability and robustness of the unhealthy robots under the premise of minor changes in the original framework. At the same time, the insertion construction method was introduced to make the robots in the formation more efficient. This enables a better response to changes in the environment.

(4) In the simulations and experiments, through experimental verification using the MATLAB simulation platform and the ROS robot platform, it was verified that the method proposed in this paper can ensure that the robot can form the required formations according to the optimal distribution. In the designated area, the structural redundancy and internal carrying capacity of the multi-robot system were increased, and the formation was dynamically switched according to the external environment. At the same time, the health of the robots was better optimized and managed, which improves the flexibility and robustness of the formation.

In future work, formation control in more complex obstacle environments will be considered. When applied to heterogeneous groups with different performance and motion models, the method proposed in this paper will be more versatile and robust. These applications will continue to be studied in future work.

**Author Contributions:** Conceptualization, K.C. and S.G.; Methodology, Y.C. and S.G.; Resources, H.D.; Software, K.C. and H.Z.; Supervision, S.G.; Writing—original draft, K.C. All authors have read and agreed to the published version of the manuscript.

**Funding:** This research was funded by Shaanxi International Science and Technology Cooperation Project grant number [2019KW-014] and Xi'an Technological University abroad funding grant number [2019001].

**Institutional Review Board Statement:** Not applicable.

**Informed Consent Statement:** Not applicable.

**Data Availability Statement:** Not applicable.

**Conflicts of Interest:** The authors declare no conflict of interest.

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
