# Peer review of "Multi-Robot Formation Control Based on CVT Algorithm and Health Optimization Management"

_applsci, doi:10.3390/app12020755_

Round 1
Reviewer 1 Report
I wish your team coninued success.
Author Response
Thank you for your review!
Reviewer 2 Report
The paper deals with a very interesting research topic, namely multi-robot training control. The abstract of the paper contains very long sentences. I think a few points are missing at the end of the sentences and words written in a sentence with capital letters, such as Form. I recommend the authors to check carefully. In the introductory part, references [1-3] are formatted differently from the rest, ie they are superscript. Likewise 9, 14. It is good that the introductory part ends with the presentation of what the paper contains.
The second part, "Multi robot formation control", contains theoretical foundations. It is not clear to me whether they represent the authors' contribution. Starting with paragraph 2.3, the authors present the formulation of the control algorithm. The kinematic model of the robot is often used.
Do I assume that wheel 3 is a self-steering one?
Part 3 presents problems regarding health optimization management. The experiments and simulations in Matlab from part 4 are very interesting. An important contribution of the work is the experimental validation presented in part 5 of the paper.
The paper is well written, and makes important contributions in the field. I congratulate the authors for their contribution.
Author Response
Paper ID: applsci-1471783
Revision according to the comments from Reviewer
- The paper deals with a very interesting research topic, namely multi-robot training control. The abstract of the paper contains very long sentences. I think a few points are missing at the end of the sentences and words written in a sentence with capital letters, such as Form. I recommend the authors to check carefully. In the introductory part, references [1-3] are formatted differently from the rest, ie they are superscript. Likewise 9, 14. It is good that the introductory part ends with the presentation of what the paper contains.
Reply: Thanks for your careful review. After careful examination of the abstract, some incorrect capitalization was corrected. Regarding the citation format of references, the author has made corrections in accordance with the citation format of the latest article in the journal, and has made yellow high light in the article.
- The second part, "Multi robot formation control", contains theoretical foundations. It is not clear to me whether they represent the authors' contribution. Starting with paragraph 2.3, the authors present the formulation of the control algorithm.
Reply: Thanks to the reviewers for their careful review, the second part of the basic theory of CVT and the author's previous contributions when applying CVT to multi-robot control. The application of CVT to the ROS multi-robot formation control algorithm, the optimization of the cost function, and the analysis of stability are the author's contributions in this article.
- The kinematic model of the robot is often used. Do I assume that wheel 3 is a self-steering one?
Reply: The robot kinematics model used in this article is based on a differential two-wheeled robot, and the third wheel is a self-steering wheel.
- Part 3 presents problems regarding health optimization management. The experiments and simulations in Matlab from part 4 are very interesting. An important contribution of the work is the experimental validation presented in part 5 of the paper.
The paper is well written, and makes important contributions in the field. I congratulate the authors for their contribution.
Reply: Thanks for your review.

Reviewer 3 Report
I found the paper very hard to understand, probably because it would need a substantial correction of the English language, also because I might not be a specialist of the domain, or the paper lacks explanations of important concepts.
Among others, here some points that I did not understand and probably prevented to understand the paper :
L127
Explain better what is a generation point
L129-L130
Tell more about what is the cost function, what do you call "the system"
L131
What do you mean by density?
L138 vs L140
What is p_i, q, which one is the position of the robot or the position of the wireless sensor?
Author Response
Paper ID: applsci-1471783
Revision according to the comments from Reviewer
I found the paper very hard to understand, probably because it would need a substantial correction of the English language, also because I might not be a specialist of the domain, or the paper lacks explanations of important concepts.
Among others, here some points that I did not understand and probably prevented to understand the paper :
Reply: Thank you again for your suggestions, the author has carefully checked all the grammar and English writing in the text and made amendments.
L127. Explain better what is a generation point
Reply: When generating point pi, it refers to the point where the Voronoi cell Vi is generated, that is, all other points q in the Voronoi cell Vi generated from this point are closer to the generating point than to the generating points in other cells. The specific mathematical form is shown in formula (1). For more detailed description and legend, see L159-L163 and Figure 1. When the subsequent application is used in the simulation and experiment of the robot generating formation, the generating point also refers to the current position of the robot, because the Voronoi cell is generated according to the current position of the robot. The definition of the Voronoi tessellations algorithm is to divide the area containing n generation points into n cells.
L129-L130. Tell more about what is the cost function, what do you call "the system"
Reply: The cost function refers to the cost required for the entire swarm of multiple robots to form a specified formation at a specified position. It is composed of the product of the distance between each robot and the target point and the density function of the target point. The mathematical formula applied in the formation of multi-robots is shown in formula (14). See L215-L221 for specific description. Physically, it can be understood as the energy required to reach the desired location.
The system refers to a group composed of multiple robots, which is a multi-robot system.
L131. What do you mean by density?
The density function ρ(q) refers to the function of the density value at each point in the current area Q. In this article, it can be understood as the degree of interest of each point in the area. The greater the degree of interest, the density function the larger the value, the smaller the opposite. The specific explanation of the density function is explained in detail at L167~L169. Formula (26) and Figure 5(a), Formula (27) and Figure 6(a) are two different density function forms.
L138 vs L140. What is p_i, q, which one is the position of the robot or the position of the wireless sensor?
Reply: pi is the position of the i-th robot, which is also used to indicate the i-th robot. q refers to any point in the entire area Q, which can be understood as any possible point of interest for the robot in the area. The Voronoi area of each robot is composed of a certain number of arbitrary points p. It is not clearly stated here, and has been changed in the text L155.

Reviewer 4 Report
In this paper, a multi-robot formation control and health optimization management method based on CVT algorithm is proposed, which has prominent advantages compared with the traditional multi-robot formation algorithm. Firstly, the algorithm converges the robot to the centroid of each Voronoi cell, which can avoid collision to the greatest extent; Secondly, the algorithm is easier to repair or reconfigure the system to ensure the execution of the task. This loosely coupled decentralized configuration greatly enhances the robustness and stability of the system; Next, the formation health optimization management algorithm is used to make the system better deal with unforeseen damage such as mechanical failure and emergencies, so as to improve the robustness of the formation when performing tasks in different scenarios; Finally, in the experimental part, by the experimental verification on MATLAB simulation platform and ROS robot platform, it is verified that the method proposed in this paper can ensure that the robot can form the desired formation according to the optimal distribution in the specified area, increase the structural redundancy and internal bearing capacity of multi-robot system, it will dynamically switch the formation according to the external environment; At the same time, the robot health is better optimized and managed, which improves the flexibility and robustness of the formation.
The article would be appropriate to explicitly indicate the scientific benefits of the proposed multi-robot formation control and health Optimization Management Method Based on Cvt Algorithm is proposed, which has prominent advantages compared with the traditional multi-robot formation algorithm.
In the present article would be appropriate to clarify how experimental identifications are used in practice.
More literary resources could be used in the present article.
The work would be appropriate to provide a more detailed literary overview of the current state in the issue using a larger number of literary resources.
Text Description Used in Pictures no. 4 to 18 are illegible in the article.
The Introduction section should contain the actuality of the problem, the current research revision in this subject area with the allocation of unsolved parts of the general problem, and finally, formulation of the research goal.
The conclusions section is very short too. This section does not reflect the brief content of the manuscript including an analysis of the obtained results. So, this section should be extended too.
Author Response
Paper ID: applsci-1471783
Revision according to the comments from Reviewer
- Text Description Used in Pictures no. 4 to 18 are illegible in the article.
Reply: Thank you for your reminder. Because the automatic compression function of the document makes the picture blurry, the author has replaced the picture with unclear text description in the picture in the text. It should be the problem of the template. We have contacted the editor.
- Introduction section should contain the actuality of the problem, the current research revision in this subject area with the allocation of unsolved parts of the general problem, and finally, formulation of the research goal.
Reply: Thank you very much for your careful suggestions, the author has re-adjusted the introduction, first introduced the current research status of formation technology, and then proposed that the formation control should be extended to a more efficient dynamic mode, that is, to allow the robot to maintain Under the premise of the corresponding formation, weaken the rigidity of the formation as much as possible. Then introduced the current research status of the formation algorithm CVT in this paper. Next, the health issues that are less considered in formation control are clarified, and the research and contribution of the combination of health management and CVT are introduced, and the goals of this article are put forward based on the existing issues of the contribution. Finally, it summarizes the contribution and work of this article.
- The conclusions section is very short too. This section does not reflect the brief content of the manuscript including an analysis of the obtained results. So, this section should be extended too.
Reply: Thanks to the reviewers for the careful review. The author has rewritten the conclusions and prospects. From contribution to simulation and experiment, the contribution and work of this article are presented separately.

Round 2
Reviewer 3 Report
Thank you for the improvements of the paper and the additional explanations. Unless I missed them, I think all these explanations could be directly added inside the paper.
I am not a native English speaker but to me, there are still quite a few sentences where the verb or pronouns seem missing and some others that are probably too long and make the paper hard to follow at times (especially in the abstract and introduction). I would strongly recommend to check that with an English specialist.